# CROSS-MODAL SELF-SUPERVISED LEARNING WITH EFFECTIVE CONTRASTIVE UNITS FOR POINT CLOUDS

## ABSTRACT

3D perception in LiDAR point clouds is crucial for an autonomous driving vehicle to properly act in 3D environment. However, manually labeling point clouds is hard and costly. There has been a growing interest in self-supervised pre-training of 3D perception models. Following the success of contrastive learning in images, current methods mostly conduct contrastive pre-training on point clouds only. Yet a self-driving vehicle is typically supplied with multiple sensors including cameras and LiDAR. In this context, we systematically study single modality, cross-modality, and multi-modality for contrastive learning of point clouds, and find that cross-modality wins over other alternatives. In addition, considering the huge difference between the training sources in 2D images and 3D point clouds, it remains unclear how to design more effective contrastive units for LiDAR. We therefore propose the instance-aware and similarity-balanced contrastive units that are tailored for self-driving point clouds. Extensive experiments reveal that our approach achieves remarkable performance gains over various point cloud models across the downstream perception tasks of LiDAR based 3D object detection and 3D semantic segmentation on the four popular benchmarks including Waymo Open Dataset, nuScenes, SemanticKITTI and ONCE.

## 1 INTRODUCTION

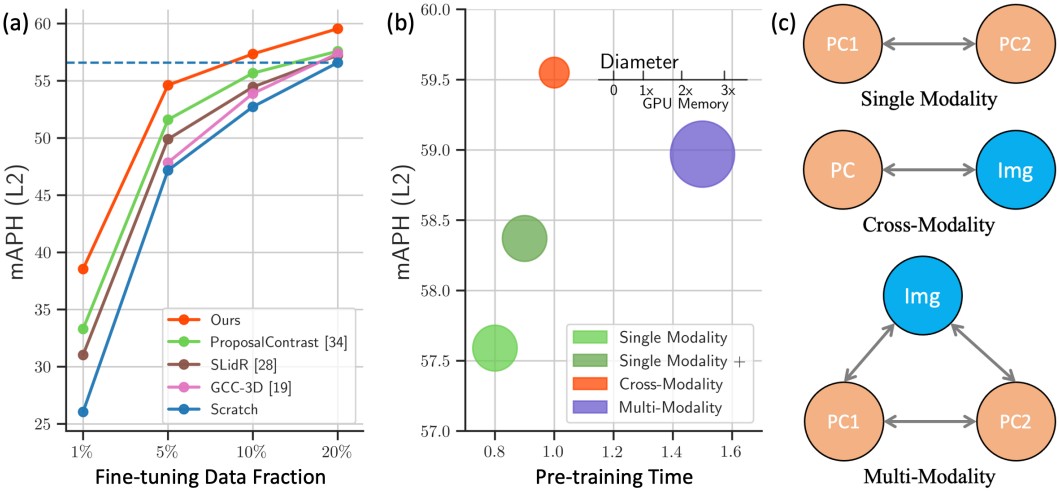

Figure 1: (a) Our approach achieves consistent and significant performance gains compared to training from scratch and other state-of-the-art self-supervised point cloud learning methods across different fractions of fine-tuning data on Waymo Open Dataset. (b) Our comprehensive modality study finds that cross-modality (ours) is superior to single modality (and its enhanced version +) and multi-modality in terms of downstream performance and memory consumption of GPU (proportional to bubble area), while requiring moderate pre-training time. (c) Illustration of single modality, cross-modality, and multi-modality for contrastive learning of LiDAR point clouds. PC1 and PC2 denote two independently augmented point clouds.

3D perception is a pivotal module of an autonomous driving vehicle as it provides the fundamental information to subsequent onboard modules ranging from prediction to planning (Guo et al., 2020; Arnold et al., 2019; Sun et al., 2020a). LiDAR is one of the most commonly utilized sensor that a self-driving system relies on to perceive its neighboring environment in 3D (Arnold et al., 2019). However, annotating LiDAR point clouds is notoriously difficult, error-prone, and time-consuming. For instance, it costs around 4.5 hours to label a single tile in SemanticKITTI (Behley et al., 2019). Recently, there has been growing attention in making use of self-supervised learning (SSL) to alleviate the laborious human labeling efforts, and at the same time, to harvest the vast amount of data continuously collected by the world-wide self-driving fleets. However, 3D SSL is still under explored compared to the well-developed family of 2D SSL methods (Chen et al., 2021; He et al., 2020; Xie et al., 2022; He et al., 2022; Chen et al., 2020).

As pioneers in 3D SSL, DepthContrast (Zhang et al., 2021) conducts contrastive pre-training by using the holistic point cloud as a contrastive unit at the scene-level, while PointContrast (Xie et al., 2020) performs point-level comparisons in two transformed point clouds with different views to capture dense information at the point-level. Such methods are designed for indoor settings captured by hundreds of scans from diverse positions per scene with limited occlusion. In contrast, LiDAR point clouds in autonomous driving capture large-scale outdoor scenes with restricted viewing angles and strong occlusions. Most LiDAR point clouds are very similar to each other from the scene-level perspective as a result of the limited diversity in street views. These differences make such scene-level 3D SSL methods incompatible with self-driving point clouds. Recently, GCC-3D (Liang et al., 2021) and ProposalContrast (Yin et al., 2022) propose to generate more fine-grained contrastive units in the region-level for LiDAR. They leverage preliminary geometric cues to drive contrastive pre-training. However, our experiments reveal that using low-level geometry makes the self-supervised objective easy to overfit and leads to the sub-optimal performance in downstream tasks.

Another track is to perform contrastive learning across images and point clouds. Pri3D (Hou et al., 2021) and PPKT (Liu et al., 2021) take the first step in exploring pixel-point correspondence for indoor point clouds. SLidR (Sautier et al., 2022) uses LiDAR point clouds and synchronized images to carry out contrastive learning, where superpixels are used to group local pixels as contrastive units. However, superpixels tend to over-segment an object into small fragments, leading to numerous false negative pairs and imbalanced sampling in the contrastive objective. Our experiments show that the pre-trained weights provided by SLidR deliver on par or even deteriorated results compared to the randomly initialized weights when fine-tuning on the downstream (large-scale annotated) datasets.

In light of the above observations, we seek to answer two fundamental research questions for LiDAR based 3D SSL: (1) *which modalities are better suited for contrastive learning of point clouds*, and (2) *how to design more effective contrastive units in self-driving scenarios*.

First, an autonomous vehicle is typically equipped with a sensor suite including cameras and LiDAR (Liu et al., 2023), offering three possible modalities to perform contrastive pre-training on: (i) single modality with point cloud only, (ii) cross-modality on image and point cloud, and (iii) multi-modality by combining (i) and (ii), as depicted in Figure 1(c). We find that *the cross-modality wins over the other two alternatives* in terms of both pre-training efficiency and downstream improvement, as shown in Figure 1(a-b). Specifically, we show that the contrastive learning on point cloud only is prone to overfit to the pre-training objective, while the multi-modality induces tremendous extra memory and computational costs yet brings no additional performance gains.

Second, a huge discrepancy exists in the training sources between 2D and 3D SSL. ImageNet (Deng et al., 2009) is the de facto training data for 2D SSL, and it is essentially a curated dataset that is instance-concentrated and class-balanced. On the contrary, real-world driving data is naturally collected at the scene-level consisting of an imbalanced compound of multiple instances and vast background with no specific focus. Inspired by this contrast, we devise *instance-aware and similarity-balanced contrastive units* in 3D SSL to approximate the counterpart in 2D SSL. In practice, we sample the initial contrastive units uniformly in a point cloud to ensure a thorough coverage of the scene. An unsupervised geometry clustering method is then introduced to merge and grow a part of the initial units into instance clusters to create the instance-aware contrastive units. As demonstrated in Figure 3(a), we can discover a rich set of foreground instances such as vehicles and pedestrians via the clustering. For the remaining initial units, a large portion are similar and monotonous from the wide-open background such as vegetation and buildings. To better balance the contrastive objective, we develop a similarity-balanced sampling to rule out the semantically similar units.

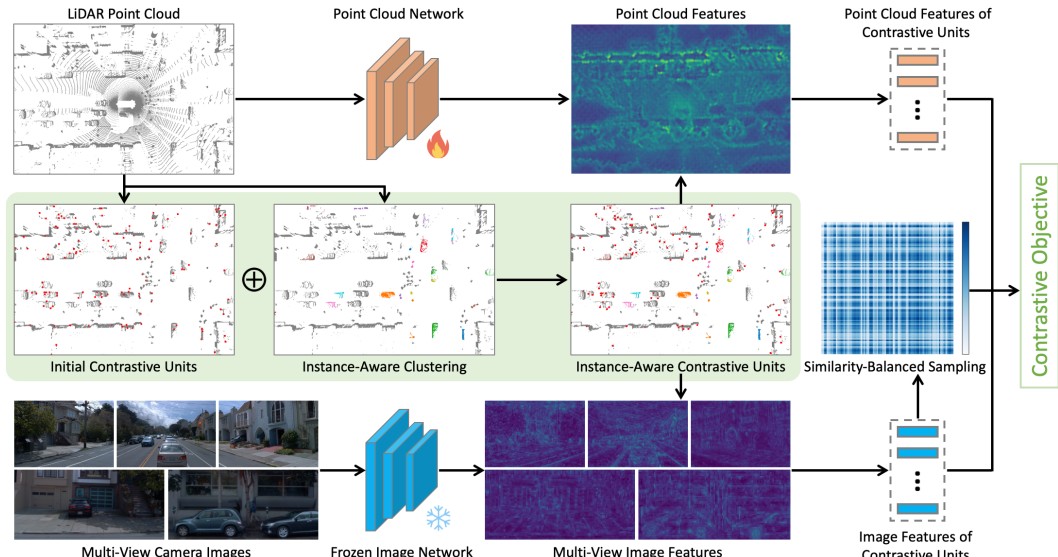

Figure 2: Overview of the proposed cross-modal contrastive pre-training framework. We uniformly sample initial contrastive units to maximally cover the point cloud scene. An unsupervised geometry clustering is introduced to generate the instance-aware contrastive units. Leveraging on the image features that are self-supervised pre-trained with rich semantics, we develop the similarity-balanced sampling to balance the contrastive objective by ruling out those units that are semantically close.

Our main contributions are summarized as follows. (1) To our knowledge, this work provides the first comprehensive study in term of modality for contrastive learning of point clouds in self-driving scenarios. Our findings show that the cross-modal learning across images and point clouds performs the best in pre-training efficiency and accuracy improvement for downstream tasks. (2) We propose the instance-aware and similarity-balanced contrastive units such that contrastive pre-training can be conducted at the instance-level with more balanced sampling. (3) Experiments reveal that our approach achieves superior performance gains on multiple downstream tasks, as demonstrated in our extensive evaluations. For instance, our pre-trained weights boost the training-from-scratch performance by 2.96% L2 mAPH on Waymo Open Dataset, exceeding the previous best result (Yin et al., 2022) by 1.91%. Our code and model will be released.

## 2 METHOD

In this section, we detail the proposed instance-aware and similarity-balanced contrastive units in cross-modal 3D SSL. We start by introducing the point cloud and image feature representations, and then describe the design of our contrastive units including instance-aware clustering and similarity-balanced sampling. Finally, we present our contrastive pre-training objective.

### 2.1 FEATURE REPRESENTATIONS

**Point Cloud Feature Representation.** Data augmentation is important to contrastive learning since it increases the difficulty of self-supervised learning, alleviates overfitting, and encourages the pre-trained weights to learn invariant features. Given a point cloud $\mathcal{P} \in \mathbb{R}^{N \times 3}$ with $N$ points, we first apply a set of transformations $\mathcal{T}$ to $\mathcal{P}$, resulting in the augmented point cloud $\mathcal{T}(\mathcal{P})$. In this paper, we use rotation, scaling, and random flipping as the augmentation set.

We denote $F_{\text{point}}$ as a point cloud network to be self-supervised pre-trained. It is used to generate the point cloud feature $P = F_{\text{point}}(\mathcal{T}(\mathcal{P}))$, where $P \in \mathbb{R}^{N \times C}$ and $C$ is the feature dimension. The goal of pre-training is to enable $F_{\text{point}}$ to learn the high-level semantics that are essential for the downstream perception tasks, but with no data labeling. Our approach is versatile to various network architectures, including the point, voxel or pillar based models.

**Image Feature Representation.** Along with LiDAR point clouds, the synchronized multi-view images in a self-driving vehicle provide extra visual information. Built upon the large-scale and well-established image datasets such as ImageNet (Deng et al., 2009), current self-supervised pre-trained networks such as MoCo (Chen et al., 2021) and SimCLR (Chen et al., 2020) offer high-quality image features with rich semantics. We therefore take advantage of such a frozen pre-trained network as the image encoder, which brings the following three benefits. First, leveraging on the success of 2D SSL, the image features learned with high-level semantics can guide the contrastive pre-training toward the high-level understanding beyond the low-level point cloud statistics. Second, the image features involving visual texture and context provide complementary information in addition to the geometric cues from point clouds. Third, the image features are frozen and utilized as "anchors" to prevent contrastive learning from overfitting. As demonstrated in our experiments, only using the point cloud features tends to lead to a "shortcut" of geometry to fulfill the pre-training objective and lack of the desired understanding in semantics.

For a point cloud $\mathcal{P}$, which is paired with $M$ synchronized images $\{\mathcal{I}_i \in \mathbb{R}^{H \times W \times 3}, i = 1, \ldots, M\}$, we use a frozen pre-trained image network $F_{\text{image}}$ to generate each image feature as $I_i = F_{\text{image}}(\mathcal{I}_i)$, where $I_i \in \mathbb{R}^{H' \times W' \times C'}$ and $H', W', C'$ denote the feature map dimensions. In our implementation, we adopt multi-scale image features from different abstraction levels as the final representation, by upsampling and concatenating feature maps from multiple resolutions. This is found to be beneficial for the downstream 3D detection and segmentation tasks in point clouds.

**Correspondence.** It is straightforward to set up the correspondence between image features and point cloud features in self-driving data. With the available calibration parameters between cameras and LiDAR, we project 3D point coordinates into 2D pixel coordinates to form the correspondence. When sampling points for initial contrastive units, we only consider the points that can be projected into at least one camera canvas. As for those points that can be projected into multiple cameras, we simply use average pooling of their corresponding features to obtain the final image representation.

## 2.2 CONTRASTIVE UNITS

**Ground Removal.** For LiDAR point clouds captured in autonomous driving, a great deal of points are collected on the ground. Sampling such points results in uninformative contrastive units in the pre-training objective, hindering the learning of true foreground objects that are more relevant for the downstream tasks. Thus, we apply a simple unsupervised ground segmentation algorithm (Himmelsbach et al., 2010) to identify and remove the LiDAR returns from ground. As shown in Figure 2, ground removal provides a more effective sampling space to generate contrastive units.

**Initial Contrastive Units.** We start from sampling individual points in the ground-removed point cloud as the initial contrastive units. Due to the scanning mechanism of LiDAR, point clouds are extremely uneven, i.e., the point density close to the ego-vehicle is tremendously higher than that far away. To initially acquire a thorough coverage of the entire scene, we utilize the farthest point sampling (FPS). In addition, inspired by recent perception works in bird's eye view (BEV) (Li et al., 2022b), we ignore the height dimension in FPS. To supply an initial unit with more context, we further sample and aggregate the features from its neighboring points, which can be sampled by either $K$ nearest neighbor points or all points inside the pillar centered at the initial unit. Given the point cloud features or image features of an initial unit and its contextual points, we apply average pooling to get the corresponding representations of the unit.

**Instance-Aware Clustering.** Though the initial contrastive units are designed to maximally cover a scene, one foreground instance such as a vehicle can be segmented into several different units, as can be seen in Figure 2. This results in undesirable negative pairs in the contrastive objective and is detrimental to the learning of semantics at the object or instance level. Fortunately, unlike images, point clouds possess accurate geometric measurements, making it possible to discover instances in an unsupervised manner. Here we make use of a simple geometry clustering algorithm (Klasing et al., 2008) after sampling the initial contrastive units using FPS, which employs a k-d tree to cluster all neighboring points within a radius as one instance in the range image. Here k-d tree is employed to gradually refine the discovered clusters, and the range image is a 2D representation of the point cloud from range view (Wang et al., 2020). We then filter out the clusters with anomalous sizes or aspect ratios (see more details in the supplementary material). As illustrated in Figure 3(a), we discover plenty of clusters or instances with meaningful semantics.

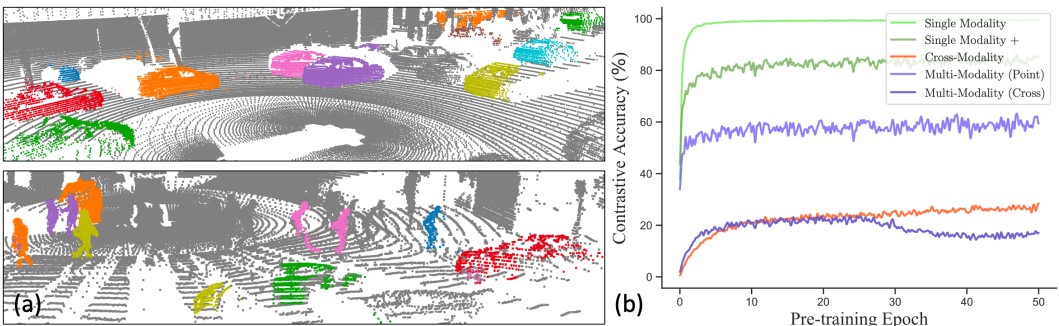

Figure 3: (a) Illustration of the instances such as vehicles and pedestrians discovered by the unsupervised clustering. Note that some instances are missing due to the imperfection of the simple rule based clustering. (b) Comparison of the contrastive accuracy of different modalities. If the similarity of a contrastive unit with its positive sample is higher than those with all negative samples, it is marked as a correct contrastive classification.

For the initial contrastive units that do not fall into any of the filtered clusters, their feature representations remain the same. As for the ones that fall into the same cluster, we merge them into a single unit, and then apply average pooling on their corresponding features to obtain the instance-level representation of the merged unit. In this way, we are able to substantially reduce the false negative pairs that are initially sampled from same instances, and meanwhile, to promote the contrastive units from initial points with relatively limited neighboring context to be instance-aware.

**Similarity-Balanced Sampling.** A lightweight multi-layer perceptron is applied as the projection head to map the image features $\{I_i\}$ and point cloud features $\{P_i\}$ of the instance-aware contrastive units to the final representations $\{\tilde{I}_i\}$ and $\{\tilde{P}_i\}$ to compute the contrastive objective. A straightforward way to conduct the contrastive learning is to exploit the corresponding cross-modal features from the same unit as a positive pair and all different units as negative pairs. However, due to the extreme foreground-background imbalance in LiDAR point clouds, numerous semantically similar units can be treated as negative pairs in the contrastive objective. For instance, if a unit is sampled from the vegetation, then other vegetation units with similar semantics would constitute a great portion of the negative pairs, as shown in Figure 2, misleading the contrastive pre-training. This is an inherent difficulty for LiDAR as self-driving point clouds are dominated by vast background.

Fortunately, leveraging the frozen image network pre-trained with rich semantics, we can take advantage of the similarity of image features to reflect how semantically close any two contrastive units are. Given two units in a point cloud, we denote their corresponding image features as $\tilde{I}_i$ and $\tilde{I}_j$, and use the cosine similarity $s_{ij} = \left\langle \frac{\tilde{I}_i}{||\tilde{I}_i||_2}, \frac{\tilde{I}_j}{||\tilde{I}_j||_2} \right\rangle$ to measure their semantic similarity, as visualized in Figure 2. Based on this measurement, we propose the similarity-balanced negative sampling strategy. Given the $i$-th unit, we measure its similarity to all other units, and keep the least similar $L$ units to form the negative pairs involved in the contrastive objective. Let the $L$-th least similarity be $s_{iL}$, then the similarity-balanced negative set for the $i$-th unit is $\mathcal{S}_i = \{j \mid s_{ij} < s_{iL}, j = 1, ..., B, j \neq i\}$, where $B$ is the number of instance-aware contrastive units. After excluding the negative pairs with high similarity in semantics, we obtain a more expressive negative set for each unit.

## 2.3 CONTRASTIVE OBJECTIVE

Based on the instance-aware and similarity-balanced contrastive units, we compute the contrastive objective by InfoNCE (Chen et al., 2020) on both image features and point cloud features of each unit for pre-training. The overall objective can be formalized as image-point cloud feature matching:

$$\mathcal{L} = -\frac{1}{2B} \sum_{i=1}^{B} \log \left[ \frac{e^{(\langle \tilde{I}_i, \tilde{P}_i \rangle / \tau)}}{\sum_{j \in \mathcal{S}_i} e^{(\langle \tilde{I}_i, \tilde{P}_j \rangle / \tau)} + e^{(\langle \tilde{I}_i, \tilde{P}_i \rangle / \tau)}} \right] - \frac{1}{2B} \sum_{i=1}^{B} \log \left[ \frac{e^{(\langle \tilde{P}_i, \tilde{I}_i \rangle / \tau)}}{\sum_{j \in \mathcal{S}_i} e^{(\langle \tilde{P}_i, \tilde{I}_j \rangle / \tau)} + e^{(\langle \tilde{P}_i, \tilde{I}_i \rangle / \tau)}} \right]$$

where $\tau$ is the temperature and $B$ is the number of contrastive units after instance-aware clustering.

| Pre-training | Performance Gain | Overall mAP/mAPH | Vehicle AP/APH | Pedestrian AP/APH | Cyclist AP/APH |
|---|---|---|---|---|---|
| Scratch* | - | 59.14/55.25 | - | - | - |
| PointContrast* (Xie et al., 2020) | 0.90/1.06 | 60.04/56.31 | - | - | - |
| GCC-3D* (Liang et al., 2021) | 2.44/2.14 | 61.58/57.39 | - | - | - |
| Scratch | - | 60.74/56.59 | 62.03/61.46 | 61.70/51.68 | 58.49/56.63 |
| PPKT (Liu et al., 2021) | 0.53/0.51 | 61.27/57.10 | 62.62/62.09 | 62.24/52.17 | 58.95/57.04 |
| SLidR (Sautier et al., 2022) | 0.66/0.64 | 61.40/57.23 | 62.40/61.87 | 62.49/52.20 | 59.30/57.64 |
| SegContrast (Nunes et al., 2022) | 0.54/0.49 | 61.28/57.08 | 62.44/61.90 | 62.39/52.10 | 59.00/57.24 |
| ProposalContrast (Yin et al., 2022) | 0.88/1.00 | 61.62/57.59 | 63.42/62.86 | 62.38/52.75 | 59.07/57.17 |
| Ours | **2.73/2.96** | **63.47/59.55** | **64.22/63.67** | **64.69/55.33** | **61.49/59.66** |

Table 1: Comparison of 3D object detection based on CenterPoint-Pillar. We report L2 AP and APH on the validation set of WOD. * denotes the results from (Liang et al., 2021).

| Pre-training | Performance Gain | Overall mAP/mAPH | Vehicle AP/APH | Pedestrian AP/APH | Cyclist AP/APH |
|---|---|---|---|---|---|
| Scratch* | - | 63.46/60.95 | 61.81/61.30 | 63.62/57.79 | 64.96/63.77 |
| GCC-3D* (Liang et al., 2021) | 1.83/1.84 | 65.29/62.79 | 63.97/63.47 | 64.23/58.47 | 67.68/66.44 |
| Scratch | - | 65.42/62.98 | 63.82/63.33 | 64.85/59.22 | 67.58/66.38 |
| PPKT (Liu et al., 2021) | 1.18/1.14 | 66.59/64.12 | 63.53/63.02 | 64.74/58.84 | 67.01/65.85 |
| SLidR (Sautier et al., 2022) | 0.69/0.67 | 66.11/63.65 | 64.34/63.84 | 66.10/60.45 | 67.87/66.68 |
| ProposalContrast (Yin et al., 2022) | 1.01/0.93 | 66.43/63.91 | 64.65/64.13 | 66.04/60.23 | 68.59/67.37 |
| Ours | **1.63/1.58** | **67.05/64.56** | **65.29/64.78** | **67.28/61.50** | **68.58/67.41** |

Table 2: Comparison of 3D object detection based on CenterPoint-Voxel. We report L2 AP and APH on the validation set of WOD. * denotes the results from (Liang et al., 2021).

# 3 EXPERIMENTS

We conduct extensive experiments on four datasets including Waymo Open Dataset (WOD) (Sun et al., 2020b), nuScenes (Caesar et al., 2020), SemanticKITTI (Behley et al., 2019), and ONCE (Mao et al., 2021). Our approach is applicable to various point cloud models. We select three representative networks in our experiments for fair comparison with previous works: CenterPoint (both pillar and voxel versions) (Yin et al., 2021) and MinkowskiNet (Xie et al., 2020). As for the image based network, we use the self-supervised pre-trained ResNet50 (Chen et al., 2021) to extract image features. We provide the dataset and implementation details in the supplementary materials.

## 3.1 COMPARISON WITH STATE-OF-THE-ART METHODS

**Comparison on WOD.** We first use CenterPoint-Pillar and follow the standard fine-tuning protocol using 30 epochs and 20% of training samples of WOD. As shown in Table 1, our approach achieves the most significant performance gain compared to training from scratch, even though GCC-3D is built upon a lower baseline (relatively easier to produce a larger gain on a lower baseline). As expected, the overall performance of point-level pre-training in PointContrast is inferior due to the fact that its granularity (dense points as contrastive units) is not suited for self-driving point clouds. Among the cross-modal methods, our approach largely outperforms SLidR thanks to our design of contrastive units that are instance-aware and similarity-balanced. Our approach remarkably improves training from scratch by 2.73% mAP and 2.96% mAPH. In particular, we observe greater boost on pedestrians (+3.65% APH) and cyclists (+3.03% APH) compared to vehicles (+2.21% APH). In contrast, ProposalContrast receives larger improvement on vehicles (+1.40% APH) than pedestrians (+1.07% APH) and cyclists (+0.54% APH). This clearly validates the advantage of visual cues provided in image features for the contrastive learning of small objects in point clouds, and pre-training on point cloud only makes it hard to guide the learning of small objects.

We then evaluate our approach using a stronger point cloud network CenterPoint-Voxel. As shown in Table 2, we find a similar trend in comparison with other contrastive learning methods, and the proposed approach still achieves superior performance with a stronger baseline or backbone. Furthermore, we compare with the leading generative masked modeling based 3D SSL in Table 3. Our approach also compares favorably with the methods in this field.

| Pre-training | Performance Gain | Overall mAP/mAPH | Vehicle AP/APH | Pedestrian AP/APH | Cyclist AP/APH |
|---|---|---|---|---|---|
| Scratch* | - | 65.60/63.21 | 64.18/63.69 | 65.22/59.68 | 67.41/66.25 |
| BEV-MAE* (Lin & Wang, 2022) | 1.32/1.24 | 66.92/64.45 | 64.78/64.29 | 66.25/60.53 | 69.73/68.52 |
| Scratch† | - | 64.51/61.92 | 63.16/62.65 | 64.27/58.23 | 66.11/64.87 |
| Voxel-MAE† (Min et al., 2022) | 1.35/1.31 | 65.86/63.23 | 64.05/63.53 | 65.78/59.62 | 67.76/66.53 |
| MAELi† (Krispel et al., 2022) | 1.09/1.08 | 65.60/63.00 | 64.22/63.70 | 65.93/59.79 | 66.66/65.52 |
| Scratch | - | 65.42/62.98 | 63.82/63.33 | 64.85/59.22 | 67.58/66.38 |
| Ours | **1.63/1.58** | **67.05/64.56** | **65.29/64.78** | **67.28/61.50** | **68.58/67.41** |

Table 3: Comparison of 3D object detection based on CenterPoint-Voxel. We report L2 AP and APH on the validation set of WOD. * denotes the results from (Lin & Wang, 2022) and † from (Min et al., 2022; Krispel et al., 2022).

| Pre-training | mAP | NDS | mAP@1 |
|---|---|---|---|
| Scratch* | 49.60 | 60.20 | - |
| GCC-3D* (Liang et al., 2021) | $50.80_{+1.20}$ | $60.80_{+0.60}$ | - |
| Scratch | 51.34 | 61.22 | 24.66 |
| SLidR (Sautier et al., 2022) | $50.82_{-0.52}$ | $61.01_{-0.21}$ | $25.59_{+0.93}$ |
| Ours | $\mathbf{52.91_{+1.57}}$ | $\mathbf{62.65_{+1.43}}$ | $\mathbf{33.06_{+8.40}}$ |

Table 4: Comparison of 3D object detection based on CenterPoint-Pillar. We report mAP, NDS, and mAP at first epoch on the validation set of nuScenes. * denotes the results from (Liang et al., 2021).

**Comparison on nuScenes.** We pre-train CenterPoint-Pillar on nuScenes, and then fine-tune for 20 epochs using 100% labeled data under the strong setting of using 10 sweeps as input. Table 4 shows that our approach enjoys not only better final performance but also faster convergence speed. After the first fine-tuning epoch, our approach already obtains 33.06% mAP, 8.40% higher than training from scratch. As for SLidR, although it gets 0.93% mAP improvement after the first epoch, its final result is inferior to that of training from scratch. This suggests that the superpixel based contrastive units are inadequate to fully drive the learning of essential semantics for downstream tasks, and its pre-training effect would be diminished when the available fine-tuning data is relatively large.

## 3.2 MODALITY STUDY

As discussed earlier, we have three choices of modalities for contrastive pre-training on point clouds: single modality (point cloud), cross-modality (image and point cloud), and multimodality (combination of the former two). For the single modality, we compare with the best point cloud based method ProposalContrast, as well as a stronger version (single modality +) by using our contrastive units.

As shown in Table 5, cross-modality achieves the best downstream performance with moderate pre-training time and requires the least GPU memory. We find that pre-training on single modality tends to overfit at an early stage. Figure 3(b) shows that the contrastive accuracy of single modality leaps to nearly 100% after the first epoch. This indicates that point clouds provide strong hints in fitting the geometry based contrastive objective, restraining the model from learning the essential semantics. Our contrastive units help to some extent,

| Modality | mAPH | Time | Memory |
|---|---|---|---|
| Scratch | 56.59 | - | - |
| Single Modality | $57.59_{+1.00}$ | 0.8× | 1.4× |
| Single Modality + | $58.37_{+1.78}$ | 0.9× | 1.4× |
| Cross-Modality | $\mathbf{59.55_{+2.96}}$ | 1.0× | 1.0× |
| Multi-Modality | $58.97_{+2.38}$ | 1.5× | 1.9× |

Table 5: Comparison of pre-training modalities for 3D object detection based on CenterPoint-Pillar on the validation set of WOD. We report L2 mAPH, pre-training time and GPU memory.

but the overfitting (single modality +) is still obvious compared to cross-modality. As for the pre-training of multi-modality, its point cloud part or cross-modality part follows a similar trend of each individual setting, while receiving the intermediate performance, as compared in Table 5. Indeed, multi-modality is unnecessary since the frozen image features already act as "anchors", and aligning cross-modal features is a harder task. If the point cloud features of two independently augmented samples (a positive pair) are pushed close to each other, and meanwhile, they are moved toward their corresponding image features that are sufficiently close as pre-trained by 2D SSL, it is adequate to optimize the point cloud features of one sample to match to its image features.

| Pre-training | 1% | 5% | 10% | 20% |
|---|---|---|---|---|
| Scratch* | - | 44.35 | 51.14 | 55.25 |
| PointContrast* (Xie et al., 2020) | - | $44.97_{+0.62}$ | $52.35_{+1.21}$ | $56.31_{+1.06}$ |
| GCC-3D* (Liang et al., 2021) | - | $47.85_{+3.50}$ | $53.89_{+2.75}$ | $57.39_{+2.14}$ |
| Scratch | 26.05 | 47.17 | 52.73 | 56.59 |
| SLidR (Sautier et al., 2022) | $31.03_{+4.98}$ | $49.90_{+2.73}$ | $54.46_{+1.73}$ | $57.23_{+0.64}$ |
| ProposalContrast (Yin et al., 2022) | $33.30_{+7.25}$ | $51.60_{+4.43}$ | $55.67_{+2.94}$ | $57.59_{+1.00}$ |
| Ours | $\mathbf{38.55_{+12.50}}$ | $\mathbf{54.62_{+7.45}}$ | $\mathbf{57.35_{+4.62}}$ | $\mathbf{59.55_{+2.96}}$ |

Table 6: Comparison of 3D object detection using CenterPoint-Pillar with different amounts of data. We report L2 mAPH on the validation set of WOD. * denotes the results from (Liang et al., 2021).

| Method | Scratch | SLidR (Sautier et al., 2022) | ProposalContrast (Yin et al., 2022) | Ours |
|---|---|---|---|---|
| mAP | 49.40 | 49.87 | 50.87 | **52.20** |

Table 7: Comparison of 3D object detection based on CenterPoint-Pillar under the transfer learning setting. We report mAP on the validation set of ONCE.

We further quantitatively compare the feature alignment under cross-modality and multi-modality. Specifically, we compute the feature cosine similarity of a positive pair as the alignment score. By randomly sampling $1 \times 10^7$ positive pairs, we observe that the cross-modal pre-training has a much higher alignment score (0.708) than that (0.532) of the multi-modal pre-training. This again shows the advantage of cross-modality over multi-modality in contrastive learning of point clouds.

## 3.3 DATA-EFFICIENT FINE-TUNING AND TRANSFER LEARNING

**Comparison on 3D Object Detection.** We gradually increase the amount of annotated training data from 1%, 5%, 10%, to 20%, and evaluate the fine-tuning performance on WOD. As shown in Table 6, our approach exhibits greater performance gains over other methods when a small quantity of labeled data is available. For instance, when merely having 1% data, we observe a dramatic improvement of 12.50% mAPH, which substantially outperforms other methods. It is also interesting to note that with 10% data, we beat the performance of training from scratch using the standard setting of 20% data, meaning that human labeling efforts can be halved with our approach.

We next study the transfer learning capability of our approach. Specifically, we adopt CenterPoint-Pillar pre-trained on WOD, then fine-tune and evaluate on the standard training and validation sets of ONCE. As shown in Table 7, our approach achieves superior improvement (+2.80% mAP) over training from scratch, which largely outperforms SLidR (+0.47% mAP) and ProposalContrast (+1.47% mAP), suggesting the strong generalizability of our approach.

**Comparison on 3D Semantic Segmentation.** We then extend our approach to 3D semantic segmentation, where we first pre-train MinkowskiNet on nuScenes, and then fine-tune on nuScenes as well as SemanticKITTI (transfer learning evaluation), following the experimental settings in SLidR. As compared in Table 8, with 1% of labeled data, our approach achieve 8.9% and 6.2% performance gains on nuScenes and SemanticKITTI, exceeding the improvement by other methods. For this downstream task, the point cloud only based pre-training methods (PointContrast and Depth-Contrast) produce much lower improvement in comparison to the cross-modal pre-training methods (PPKT, SLidR, and ours), which indicates the benefit of visual information provided by image features in contrastive pre-training to facilitate this fine-grained point-wise perception task.

## 3.4 ABLATION STUDY

We perform various ablation experiments to understand each individual component of our approach, as shown in Table 9. We first evaluate different ways of feature aggregation for a contrastive unit as mentioned in Section 2.2, including $K$ nearest neighboring points of the unit or the points within the pillar centered at the unit. It is observed that the two ways are overall comparable, showing the flexibility of our approach. Compared to the full pre-training framework, removing either instance-aware clustering or similarity-balanced sampling results in a performance drop. Moreover, changing farthest point sampling to random sampling for initial contrastive units leads to lower performance. These ablation study results collectively validate the proposed contrastive unit design.

| Pre-training | nuScenes | Gain | SemanticKITTI | Gain |
|---|---|---|---|---|
| Scratch | 30.3 | - | 39.5 | - |
| PointContrast (Xie et al., 2020) | 32.5 | 2.2 | 41.1 | 1.6 |
| DepthContrast (Zhang et al., 2021) | 31.7 | 1.4 | 41.5 | 2.0 |
| PPKT (Liu et al., 2021) | 37.8 | 7.5 | 43.9 | 4.4 |
| SLidR (Sautier et al., 2022) | 38.3 | 8.0 | 44.6 | 5.1 |
| SegContrast (Nunes et al., 2022) | 31.9 | 1.6 | - | - |
| Ours | **39.2** | **8.9** | **45.7** | **6.2** |

Table 8: Comparison of 3D semantic segmentation based on MinkowskiNet. We report mIOU on the validation sets of nuScenes and SemanticKITTI (the latter is under the transfer learning setting).

| Pillar | Neighbor | Instance | Similarity | FPS | mAPH | mAP |
|---|---|---|---|---|---|---|
| ✓ | | ✓ | | ✓ | 59.18 | 62.99 |
| ✓ | | ✓ | ✓ | ✓ | 59.03 | 62.85 |
| | ✓ | ✓ | | ✓ | 59.17 | 63.00 |
| | ✓ | | ✓ | ✓ | 58.80 | 62.72 |
| | ✓ | ✓ | ✓ | | 58.71 | 62.62 |
| | ✓ | ✓ | ✓ | ✓ | **59.55** | **63.47** |

Table 9: Ablation study of different combinations of feature aggregation based on nearest neighbor and pillar, instance-aware clustering, similarity-balanced sampling and FPS. We report L2 mAP and mAPH using CenterPoint-Pillar on the validation set of WOD.

# 4 RELATED WORK

**Self-Supervised Learning in 2D.** Early works hinging on pretext tasks (Gidaris et al., 2018; Zhang et al., 2016) are limited to learning low-level cues. More recent contrastive learning methods like MoCo (He et al., 2020) and SimCLR (Chen et al., 2020) align the features of augmentations from the same image while pushing away other images, and achieve similar linear probing performance to fully supervised pre-training. Masked image modeling (MIM) (He et al., 2022) employs a high mask ratio to reconstruct an image in a generative way and shows promising results.

**Self-Supervised Learning in 3D.** Inspired by 2D SSL, contrastive learning and masked modeling are the two main tracks for 3D SSL. PointContrast (Xie et al., 2020) exploits point clouds from two views to build the contrastive pre-training objective. DepthContrast (Zhang et al., 2021) applies both point and voxel based backbones to extract features of each contrastive unit. Recently, GCC-3D (Liang et al., 2021) introduces a two-stage pre-training paradigm to treat a local neighborhood and motion group as the contrastive unit. In (Yin et al., 2022), ProposalContrast uses proposals and online clustering to perform contrastive pre-training. Another line is to conduct contrastive learning across images and point clouds to alleviate the limitations of geometry-only SSL. Pri3D (Hou et al., 2021) and PPKT (Liu et al., 2021) use pixel-point correspondences to build the pre-training objective. SLidR (Sautier et al., 2022) introduces superpixels to group neighboring pixels in 2D as a region-level contrastive unit. Generative masked auto-encoding has also been explored for point clouds. Occupancy-MAE (Min et al., 2022), BEV-MAE (Lin & Wang, 2022) and MAELi (Krispel et al., 2022) all utilize a masked auto-encoder in the space of BEV. GeoMAE (Tian et al., 2023) further utilizes geometry clues such as surface normal and curvature as the self-supervised objective.

# 5 CONCLUSION

We present a cross-modal self-supervised learning framework with the proposed effective contrastive units for self-driving point clouds. We provide the first comprehensive modality study of contrastive learning for LiDAR, and show that cross-modal learning performs the best for both pre-training efficiency and downstream improvement. Our contrastive units facilitate contrastive pre-training via the design of instance-aware clustering and similarity-balanced sampling. Extensive experiments reveal that our approach achieves remarkable performance gains. We hope our findings would encourage the research community on the cross-modal and more targeted designs of self-driving point clouds.

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

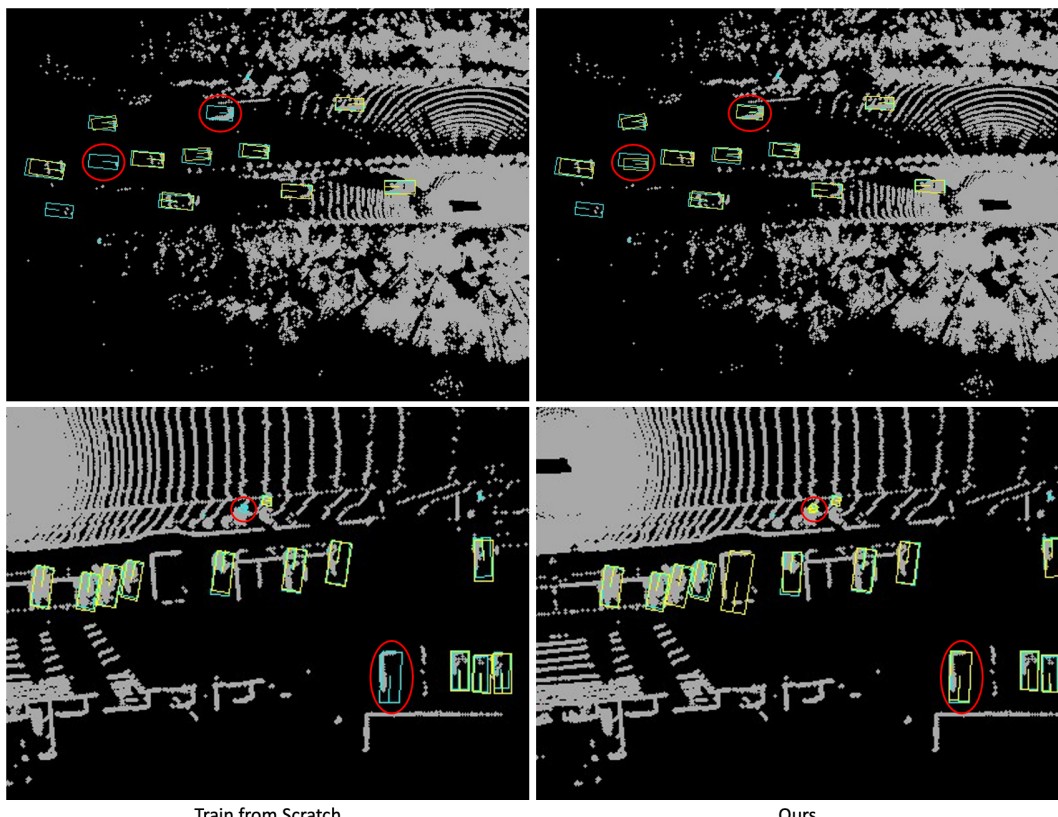

Train from Scratch                              Ours

Figure 4: Comparison of qualitative results by training from scratch (left) and ours (right). We show 3D object detection results in bird's eye view (cyan and yellow denote ground-truth and predicted boxes) on the validation set of WOD. Our pre-training benefits this downstream task in tackling more challenging cases such as the distant, occluded and small objects.

## APPENDIX

We thank the reviewers for viewing this supplementary material. In Section A, we show the qualitative results of 3D object detection with our proposed approach. Section B visualizes the similarity measurement used for similarity-balanced sampling. Section C reports more ablation studies for better understanding of our design. Section D presents the dataset and implementation details. Finally, we visualize the pre-trained point cloud features in Section E.

## A  QUALITATIVE RESULTS OF 3D OBJECT DETECTION

Our proposed cross-modal pre-training approach for LiDAR point clouds benefits downstream tasks such as 3D object detection, especially for the distant, occluded or small objects. Figure 4 demonstrates the 3D object detection results using CenterPoint-Pillar on WOD. In the left and right scenes, with our pre-trained weights, more vehicles and pedestrians can be successfully detected.

## B    SIMILARITY VISUALIZATION

Another benefit of our cross-modal paradigm is to utilize the frozen image features that are pre-trained with rich semantics to measure the similarity between contrastive units, which serves as a key in our design of similarity-balanced sampling. Here we show several scenes to demonstrate the similarity measurement. As illustrated in Figure 5, the contrastive units that are semantically close indeed share similar measurements. For instance, given the target unit from vegetation in (a), the contrastive units from vegetation own highest similarities compared to others from regions such as buildings and vehicles, similarly in (b-c).

## C    MORE ABLATION STUDIES

We provide more ablation studies to further evaluate and understand our approach. All experiments in this section are fairly compared by pre-training CenterPoint-Pillar for 20 epochs on WOD. We report L2 mAP and mAPH on the validation set of WOD.

### C.1    IMAGE FEATURE LEVELS

Given the frozen image backbone MoCoV3, we extract and combine image features from different levels. Based on three levels corresponding to the scales of 1/4 (P2), 1/8 (P3), and 1/16 (P4) of the input image size, we evaluate the three choices, namely P2, P4, and P2+P3+P4. As shown in Table 10, the image features concatenated from three levels achieve the best performance, and P4 outperforms P2 due to its high-level semantics from deep abstraction. This suggests that image features from different levels are more advantageous to cross-modal contrastive learning.

| Feature Level | P2 | P4 | P2+P3+p4 |
|---|---|---|---|
| mAP | 62.09 | 62.67 | 62.89 |
| mAPH | 58.01 | 58.69 | 59.02 |

Table 10: Comparison of different image feature levels used in our cross-modal contrastive learning.

### C.2    NUMBER OF INITIAL CONTRASTIVE UNITS

Here we study the effect of number of initial contrastive units. As shown in Table 11, the default value 2,048 achieves the best performance, either a smaller or larger value results in a performance drop. We hypothesize that a large amount of redundant units generate dense and similar samples (eventually form negative pairs) from a scene, misleading the contrastive objective, while fewer units are insufficient to construct a large contrastive pool (important for effective self-supervised learning (Chen et al., 2020; He et al., 2020)).

| # of Initial Contrastive Units | 1024 | 2048 | 4096 |
|---|---|---|---|
| mAP | 62.64 | 62.89 | 62.48 |
| mAPH | 58.80 | 59.02 | 58.54 |

Table 11: Comparison of different numbers of sampling initial contrastive units in our approach.

### C.3    NUMBER OF NEIGHBORING POINTS

In addition to the different feature aggregation ways as shown in Table 9 of the main paper, we further evaluate different numbers of points for the nearest neighboring based feature aggregation. As shown in Table 12, we observe that more neighboring points improve the performance, while the improvement margin is diminishing.

| # of Neighboring Points | 8 | 16 | 32 |
|---|---|---|---|
| mAP | 62.62 | 62.89 | 62.95 |
| mAPH | 58.81 | 59.02 | 59.11 |

Table 12: Comparison of different numbers of points in nearest neighbor based feature aggregation.

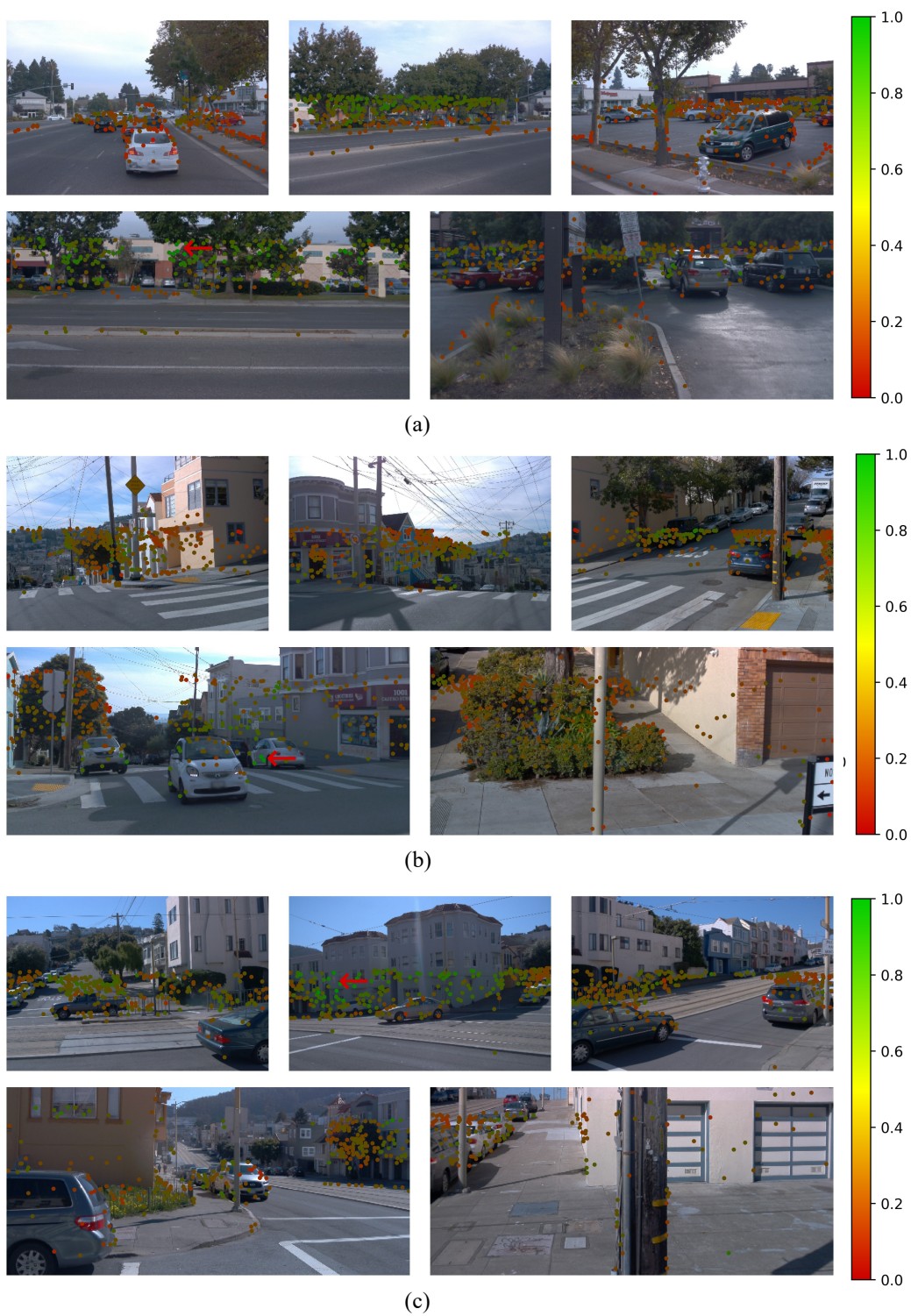

Figure 5: Visualization of similarity measurements between other contrastive units and the target unit (indicated by the red arrow) from (a) vegetation, (b) vehicle, and (c) building. Color of each dot denotes the similarity measurement from low (red) to high (green).

## C.4 IMAGE NETWORKS

By default, we use MoCoV3 (Chen et al., 2021), which is a CNN based (ResNet50) image backbone. We also experiment with the image network EsViT (Li et al., 2022a), which is based on Transformers (Swin-T). We extract and concatenate image features from three levels, and conduct the same contrastive pre-training. As shown in Table 13, EsViT also brings obvious improvement compared to training from scratch, but its improvement is inferior to MoCoV3. We suspect that the patch encoding in Transformers partially breaks the point-pixel feature correspondence because of the large patch size. Moreover, we also apply a randomly initialized ResNet50 to extract image features. As can be seen in Table 13, the randomly initialized image network only slightly improves the downstream performance compared with training from scratch. This reveals that a self-supervised pre-trained image network is crucial to the success of cross-modal contrastive pre-training.

| Image Backbone | Scratch | MoCoV3 | EsViT | Random |
|---|---|---|---|---|
| mAP | 60.74 | 62.89 | 62.51 | 61.18 |
| mAPH | 56.59 | 59.02 | 58.54 | 56.83 |

Table 13: Comparison of different image networks used in our cross-modal contrastive learning.

## C.5 GROUND REMOVAL

The ground removal is a pre-processing step for our initial contrastive unit sampling. We show the impact of this operation in Table 14. As we can see, without ground removal, there is a 0.37% mAP performance drop, indicating that sampling points from vast ground area is detrimental.

| Ground Removal | w/ | w/o |
|---|---|---|
| mAP | 62.89 | 62.52 |
| mAPH | 59.02 | 58.57 |

Table 14: Comparison of the effect of ground removal in our approach.

## D DATASET AND IMPLEMENTATION DETAILS

In this section, we provide more details regarding the four public benchmarks and our experimental implementation in addition to the descriptions in the main paper.

## D.1 DATASETS

We experiment on the four popular autonomous driving benchmarks including Waymo Open Dataset (WOD) (Sun et al., 2020b), nuScenes (Caesar et al., 2020), SemanticKITTI (Behley et al., 2019), and ONCE (Mao et al., 2021). **WOD** contains 798 scenes for training and 202 scenes for validation. We use all data in the training set for pre-training without using any labels. Both images and point clouds are collected at 10Hz synchronously, which naturally form the image and point cloud correspondence. However, the camera horizontal field of view does not cover the full 360-degree view, so we only pre-train on the points that can be mapped onto the cameras. **nuScenes** includes 700 and 150 scenes for training and validation. We adopt all training frames for pre-training with no labels used. The frequency of cameras and LiDAR are 12Hz and 20Hz, and we use the minimal timestep gap to decide the image and point cloud correspondence. **SemanticKITTI** is used to evaluate the transfer learning generalizability of our approach to domain change. We pre-train on nuScenes and conduct fine-tuning and evaluation on SemanticKITTI. **ONCE** is also utilized to show the transfer learning capability of our approach. We pre-train on WOD then fine-tune and evaluate on ONCE.

## D.2 IMPLEMENTATION

Each point cloud based network is pre-trained for 50 epochs. In CenterPoint, we use Adam (Kingma & Ba, 2015) and adopt the cosine annealing scheduler with warmup. We set the maximum learning rate as 0.003 and the batch size as 4 on each GPU. In MinkowskiNet, we follow (Sautier et al., 2022) to use SGD with an initial learning rate of 0.5 and a cosine annealing scheduler. Both pre-training

and fine-tuning are conducted on 8 NVIDIA V100 GPUs. For MoCoV3, we extract and combine image features at three levels corresponding to the scales of 1/4, 1/8, and 1/16 of the input image size. We set the temperature $\tau = 0.1$ in contrastive objective following SimCLR (Chen et al., 2020). We use consistent hyper-parameters across all datasets, including ground removal, instance-aware clustering, and similarity-balanced sampling.

For pre-training, we apply the same voxelization as in the downstream tasks. Our augmentations applied on point clouds include random flipping along $x$ and $y$ axes, random scaling with a range of $[0.8, 1.25]$, and random rotation of $[-\pi, \pi]$ along the z-axis. Color jittering is used for image augmentation. We sample 2,048 initial contrastive units in FPS. In the similarity-balanced sampling, we drop the negative pairs of a contrastive unit by the top 5% similarity measurement.

### D.3 CONTRASTIVE UNIT SAMPLING

We apply the ground removal[1] to refine the sampling space of initial contrastive units by removing the points from vast ground area. To be specific, we fit a plane in the point clouds given the height of LiDAR. As shown in Figure 2 of the main paper, the ground removal pre-processing provides a more informative sampling space.

After that, we apply the unsupervised geometry clustering[2] for the non-ground points, followed by the simple filtering based on the shape of each cluster. For the clustering, we use 0.75m as the maximum distance that two points can be grouped into a single cluster. For the filtering, we simply exclude the clusters with anomalous sizes or aspect ratios. Suppose the length, width, and height of a cluster are $(L, W, H)$, this cluster would be ruled out if any of the following conditions is satisfied: (1) $L > 20$m or $W > 20$m, (2) $H > 2.5$m or $H < 0.4$m, (3) $L/W > 10$ or $W/L > 10$, (4)$L/W < 0.1$ or $W/L < 0.1$.

### D.4 DOWNSTREAM FINE-TUNING

For the experiments of 3D object detection on WOD and nuScenes (Tables 1- 4 in the main paper), we adopt the original implementation and setting in CenterPoint (Yin et al., 2021). We fine-tune 30 epochs on WOD and 20 epochs on nuScenes using 8 NVIDIA Tesla V100 GPUs.

For the data-efficient experiment of 3D object detection on WOD (Table 6 in the main paper), we use the annotated frames proportional to the percentages, and meanwhile, sample the copy-and-paste database from the corresponding frames for a fair comparison with ProposalContrast (Yin et al., 2022) and GCC-3D (Liang et al., 2021). As for the transfer learning experiment on ONCE (Table 7), we use the standard training and validation splits defined in the official codebase[3], which includes 4,961 frames for training and 3,321 frames for validation. In practice, we fine-tune 80 epochs on ONCE using 8 NVIDIA Tesla V100 GPUs.

For the data-efficient and transfer learning experiments of 3D semantic segmentation on nuScenes and SemanticKITTI (Table 8 in the main paper), we apply the original setting in SLidR (Sautier et al., 2022), including the labeled data and the point cloud network. Specifically, we fine-tune 100 epochs on each dataset using 1 NVIDIA Tesla V100 GPU.

### D.5 FAIRNESS OF COMPARISON WITH OTHER METHODS

We strictly conduct all experiments to ensure fairness with other methods, including the amount of pre-training data, pre-training epochs, and backbone models. (1) We only use LiDAR and camera data from the annotated set in WOD and nuScenes, i.e., 158,081 and 28,130 samples, respectively. Note we do not use the unlabeled data, so our pre-training data is exactly the same as ProposalContrast, GCC-3D, SLidR, and MAE based methods. (2) We keep the same pre-training epochs (i.e., 50) for all models that are open-sourced (ProposalContrast, PPKT, SLidR, and ours) in both WOD and nuScenes. (3) For a specific network, we pre-train the same set of weights for each model, there are no additional weights for pre-training in our approach.

---

[1]`https://github.com/peiyunh/ff`
[2]`https://github.com/placeforyiming`
[3]`https://github.com/PointsCoder/Once_Benchmark`

Our implementation is built upon OpenPCDet[4] and CenterPoint[5]. We reproduce ProposalContrast (Yin et al., 2022) using the official codebase[6]. For SLidR (Sautier et al., 2022), we apply the standard CenterPoint (Yin et al., 2021) backbones while keeping other settings the same as the official implementation[7]. For the other competing methods, we report the original numbers from the corresponding papers.

# E  VISUALIZATION OF PRE-TRAINED POINT CLOUD FEATURES

Our pre-trained point cloud networks learn the semantically meaningful features, which benefits multiple downstream perception tasks. Here we show the t-SNE visualization of the pre-trained point cloud features on WOD in Figure 6, where we colorize each point with the ground-truth label (provided in 3D semantic segmentation). It is found that the features from each class group into the same cluster, such as vegetation, car, and building. In addition, we observe that the features of road are clustered into a tight region too. This is because our simple rule based ground removal is by no means perfect, some ground points are left in pre-training. As a result, the segmentation performance for the ground related class such as road in the downstream task would not be impaired.

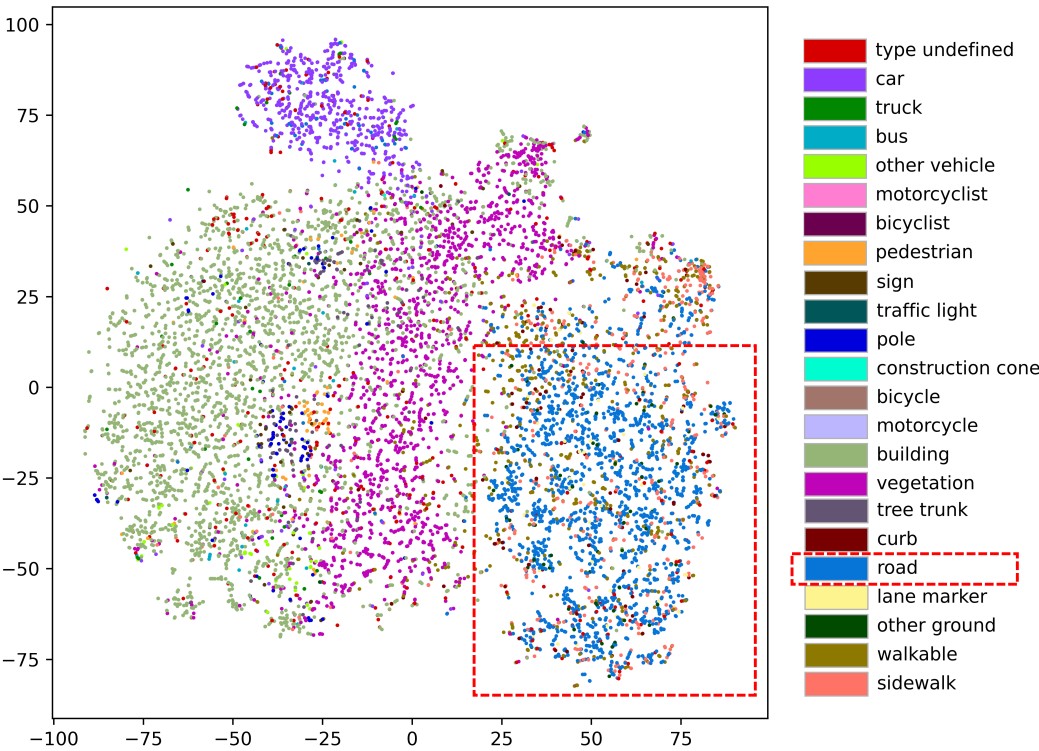

Figure 6: t-SNE visualization of point cloud features extracted by the pre-trained network with our cross-modal contrastive learning on WOD.

---

[4]https://github.com/open-mmlab/OpenPCDet

[5]https://github.com/tianweiy/CenterPoint

[6]https://github.com/yinjunbo/ProposalContrast

[7]https://github.com/valeoai/SLidR

