# OpenReview forum: "Cross-Modal Self-Supervised Learning with Effective Contrastive Units for Point Clouds"
_ICLR.cc/2024/Conference — ICLR 2024 Conference Withdrawn Submission_

### Official Review · Reviewer_djbZ · 2023-10-28

**Soundness:** 2 fair
**Presentation:** 2 fair
**Contribution:** 2 fair
**Rating:** 3
**Confidence:** 4

**Summary:**

This paper delves into the utilization of self-supervised learning (SSL) for enhancing 3D perception tasks, notably 3D object detection and LiDAR semantic segmentation. The authors categorize the prevailing techniques into three distinct clusters: 1) single modality SSL, 2) cross-modality SSL, and 3) multi-modality SSL. The authors assert a noteworthy conclusion, i.e., *“the cross-modality wins over the single modality and multi-modality”*. To verify it, the authors show empirical results along with the pre-training time of several existing SSL methods.

To augment the efficiency of SSL, the authors introduce **instance-aware and similarity-balanced contrastive units**. This allows for more nuanced, instance-level contrastive pre-training with an equilibrium in sampling. Preliminary heuristic strategies are employed to distinguish instances abundant in geometric details from LiDAR point clouds devoid of ground data. Subsequently, a multi-layer perceptron is harnessed as the projection apparatus to equate image and point features for the instance-aware contrastive units to formulate the contrastive objective.

Acknowledging the inherent long-tailed distribution in self-driving scenarios, a proposal is made by the authors to ensure a balance between the foreground and background points via similarity-balanced sampling. Several experiments are conducted on the Waymo Open Dataset, nuScenes, SemanticKITTI, and ONCE.

**Strengths:**

- This work targets the theme of self-supervised learning on point clouds, which is an interesting yet important topic.
- The proposed instance-aware and similarity-balanced contrastive learning is intuitive and very-motivated.
- The proposed approach is validated on four popular 3D perception databases.

**Weaknesses:**

- Portions of the proposed design appear derivative. A deeper exploration highlighting the unique attributes in contrast to existing methodologies is conspicuously absent.
- The paper occasionally falters in its logical progression, potentially hampering comprehension.
- The comparative analysis and literature review with contemporary and pertinent research is not adequately exhaustive.

**Questions:**

- **Q1:** In **[Sec. 2.1, Image Feature Representation]**, the justification for employing "a frozen pre-trained network as the image encoder" seems ambiguous. The SLidR paper [R1] has previously concluded similar findings. Could the authors elucidate their rationale for placing these statements at the onset of the Methodology section?

- **Q2:** Regarding **[Sec. 2.1, Correspondence]**, how does the proposed method remain functional in scenarios where potential misalignment between the camera and LiDAR ensues due to calibration errors?

- **Q3:** The ground removal and instance clustering approach mentioned in **[Sec. 2.2, Contrastive Units]** bear resemblance and similarity to techniques introduced in SegContrast [R2] and TARL [R3]. Could the authors cite and contrast these references for clarity?

- **Q4:** In **[Sec. 2.2, Similarity-Balanced Sampling]**, the concept appears to parallel the idea presented in ST-SLidR [R4], where a Semantically-Tolerant Contrastive Loss is leveraged to handle semantically close contrastive units during the cross-modality contrastive learning. How does the current approach distinguish itself in terms of novelty and efficiency?

- **Q5:** Would relocating the **[Sec. 4, Related Work]** section preceding the Methodology offer readers a more comprehensive understanding of the research backdrop?

- **Q6:** The current version seems to overlook several recent 3D SSL studies, notably [R3], [R4], [R5], [R6], and [R7]. How would the incorporation of these works influence the assertions made in this paper?

- **Q7:** Would enriching **[Sec. 5, Table 6]** with results from semi-supervised or weakly-supervised 3D object detection methodologies provide a more holistic view?

- **Q8:** The claim that *"the cross-modality is superior to the single modality and multi-modality"* could be perceived as an overstatement. Would a more nuanced conclusion be more appropriate?

- **Q9:** **[Format]**. For enhanced readability, it is suggested that table captions be positioned above the respective tables.


---

References:

- [R1] C. Sautier, et al. “Image-to-Lidar Self-Supervised Distillation for Autonomous Driving Data.” CVPR, 2022.

- [R2] L. Nunes, et al. “SegContrast: 3D Point Cloud Feature Representation Learning through Self-supervised Segment Discrimination.” IEEE Robotics and Automation Letters, 2022.

- [R3] L. Nunes, et al. “Temporal Consistent 3D LiDAR Representation Learning for Semantic Perception in Autonomous Driving.” CVPR, 2023.

- [R4] A. Mahmoud, et al. “Self-Supervised Image-to-Point Distillation via Semantically Tolerant Contrastive Loss.” CVPR, 2023.

- [R5] B. Pang, et al. “Unsupervised 3D Point Cloud Representation Learning by Triangle Constrained Contrast for Autonomous Driving.” CVPR, 2023.

- [R6] Z. Zhang, et al. “GrowSP: Unsupervised Semantic Segmentation of 3D Point Clouds.” arXiv, 2023.

- [R7] Y. Liu, et al. “Segment Any Point Cloud Sequences by Distilling Vision Foundation Models.” arXiv, 2023.

**Details Of Ethics Concerns:**

No or only a minor concern regarding the ethics.

---

### Official Review · Reviewer_uZYu · 2023-10-30

**Soundness:** 3 good
**Presentation:** 3 good
**Contribution:** 2 fair
**Rating:** 6
**Confidence:** 3

**Summary:**

In this paper, the authors proposed to improve self-supervised learning for LiDAR-based 3D perception in the autonomous driving scenario by cross-modality features from both LiDAR and images. Specifically, the proposed method consists of the following parts: 1) generate initial contrastive units from the point cloud with FPS and instance-aware clustering; 2) use the project image features (pre-trained with some self-supervised method) to obtain L least similar units in terms of image feature cosine similarity; 3) construct the contrastive learning objective via cross-modality InfoNCE loss. The authors show that the proposed method can achieve best self-supervised learning results across a number of datasets (WOD, nuScenes, ONCE) and different detection models (centerpoint-pillar, centerpoint-voxel), compared with existing single and multi-modality self-supervised learning methods. The authors also conducted empirical experiments showing that cross-modality objective is better than single- and multi-modality objective.

**Strengths:**

- The paper is overall well-written and easy to understand.
- The experiments in the paper are extensive and convincing. The authors show consistently better self-supervised learning performance on multiple datasets, detection models, and settings. As a reader, I would appreciate the effort that the authors make to carry out these experiments.
- The proposed method is overall simple and reasonable. The modality study is quite interesting and informative.
- The related work section covers quite a complete set of recent self-supervised learning works.

**Weaknesses:**

- Hard to draw conclusions from the ablation study (Table 9). The design of the ablation study looks a bit arbitrary, i.e., it is a bit hard to compare across different rows and draw conclusions because there is more than one part changed between rows. Thus it is hard to know which part plays the most significant role.
- Variance of the detection performance. Since the performance gain is relatively small for self-supervised methods, the variance of the result could lead to misleading conclusions. I wonder how larger the variance would be across different sets of runs or different sets of training samples?

Minor:
- Wrong citation for MinkowskiNet on page 6. It should be
Choy, C., Gwak, J., & Savarese, S. (2019). 4d spatio-temporal convnets: Minkowski convolutional neural networks. In Proceedings of the IEEE/CVF conference on computer vision and pattern recognition (pp. 3075-3084).

**Questions:**

See the weaknesses section.

---

### Official Review · Reviewer_BAN3 · 2023-11-01

**Soundness:** 3 good
**Presentation:** 3 good
**Contribution:** 2 fair
**Rating:** 3
**Confidence:** 4

**Summary:**

This paper presents a contrastive learning based cross-modal self-supervised learning method for point clouds in self-driving scenarios. The authors propose the instance-aware and similarity-balanced contrastive units to construct contrastive learning at the instance-level between point clouds and images. And comprehensive experiments on 3D object detection and 3D semantic segmentation tasks demonstrate the effectiveness of the proposed method.

**Strengths:**

(1) The paper proposes an instance-aware contrastive unit for cross-modal self-supervised point cloud learning in
self-driving scenarios.
(2) The organization and presentation of the paper are clear and easy to read.
(3) The experiments demonstrate the effectiveness of the proposed method.

**Weaknesses:**

(1) The main contribution of this paper is the proposed instance-aware contrastive units, and the contribution is somewhat insufficient. Basically, the instance-aware contrastive units are constructed with an instance-aware geometry clustering method of point clouds. Moreover, the motivation of using the instance-aware contrastive units is not clear. The authors map the instance-aware unit features of point clouds and image features of multi-view images to obtain the final representation for contrastive learning. In the constructed contrastive units, can we extract instance-aware features for multi-view images to form the final representation?
(2) Could you please provide more intuitions of the proposed pretrained model that can benefit the 3D object detection task? Particularly, the proposed instance-aware contrastive units is useful for 3D object detection.
(3) Ablation study analysis in this paper is insufficient. From Table. 9, it is difficult to see the advantages of introducing the instance-aware contrastive units. Moreover, the discussions on the parameters of the instance-aware clustering are missed. It could have the effects on the final self-supervised representation learning results. In addition, the discussions of the number of multi-view images are also missed in the ablation study.
(4) In table 2,  the result marked in bold (68.58) is not the best.

**Questions:**

Please see the weakness.

---

### Official Review · Reviewer_6wPD · 2023-11-01

**Soundness:** 2 fair
**Presentation:** 2 fair
**Contribution:** 2 fair
**Rating:** 3
**Confidence:** 5

**Summary:**

The paper studies self-supervised learning for 3D detection and segmentation. It investigates single-modality, cross-modality, and multi-modality for contrastive learning. The proposed algorithm utilizes existing methods to find meaningful units to construct pairs for contrastive learning. The result shows better performance on multiple benchmarks.

**Strengths:**

1. The result shows better performance on multiple benchmarks.

**Weaknesses:**

1. I think the paper is missing some important details.
    - The point cloud feature is in BEV and the image feature is in frontal view. How does the paper find correspondence?
    - The point cloud is augmented. How does the paper get the corresponding image after augmentation? For example, the image should also have the corresponding change if the point cloud is rotated or flipped.
    - How does the paper find features of instances in images?
    - What are the dimension of {$I_i$}, {$P_i$}, {$\tilde{I}_i$}, {$\tilde{P}_i$}?
    - What is $<\tilde{I}_i, \tilde{P}_i>$ in learning objective? Is it asymmetric similarity or distance? Otherwise, why does it need the second term, i.e. $<\tilde{P}_i, \tilde{I}_i>$?
1. One key approach in the paper is to find pseudo instances to contrast. How good are they? For example, can the paper show the recall of instances?
1. The definition of the modality in the paper is confusing. In Figure 1(c), to my understanding, people already defined img + PC as multi-modality. I don't quite understand why the paper needs to re-defines it as img + PC1 + PC2.
1. The paper claims to study different modalities as the main contribution. Then it should at least in all the tables categorize those existing algorithms into their types to clearly demonstrate the idea.
1. The figure 2 doesn't help to explain the idea. For example, one of the selling points is "similarity-balanced sampling", but I don't see it clearly shown in the figure.
1. The paper claims that "our approach is versatile to various network architectures, including the point, voxel or pillar based models.", but I don't see the experiment of the point-based algorithm on 3D detection, for example, PointRCNN.
1. The paper claims that "only using the point cloud features tends to lead to a shortcut of geometry to fulfill the pre-training objective", but I don't see strong evidence or ablation to support it.
1. The baselines shown in the tables are low, which may lead to unfair comparisons to other methods. For example, ProposalContrast has higher baselines and improvements in its original paper. I suggest the paper conduct experiments with stronger detectors as ProposalContrast to prove the effectiveness. Also according to the paper, it seems to be flexible about the architecture.
1. I suggest the paper show Waymo to KITTI for transfer learning as it is a more general setting.

**Questions:**

Please see the weaknesses. I will raise my score accordingly if my concerns are clarified.